# OpenReview forum: "TRACE: A Comprehensive Benchmark for Continual Learning in Large Language Models"
_ICLR.cc/2024/Conference — ICLR 2024 Conference Withdrawn Submission_

### Official Review · Reviewer_UTqq · 2023-10-25

**Soundness:** 3 good
**Presentation:** 3 good
**Contribution:** 3 good
**Rating:** 6
**Confidence:** 4

**Summary:**

The paper introduces TRACE, a new benchmark with eight diverse datasets to assess LLMs' continual learning. Tasks include domain-specific, multilingual, code generation, and mathematical reasoning tasks. The datasets are resampled with 5,000 training samples and 2,000 test samples for each task. After training on TRACE, Results show that LLMs suffer a decline in their general and instruction-following abilities. Training on reasoning tasks can offset some of these losses. Based on this, the paper proposes the Reasoning-augmented Continual Learning (RCL) method, which combines task-specific indicators with meta-rationales to prevent catastrophic forgetting in LLMs.

**Strengths:**

* The introduction of TRACE offers a comprehensive benchmark for evaluating continual learning in LLMs. TRACE encompasses a broad range of tasks, ensuring a well-rounded evaluation of LLM capabilities.

* Experiments provide valuable insights and evidence for claims in many previous papers. For example, training on reasoning tasks can mitigate the loss of general abilities.

* Designed CL metrics are reasonable and practical to evaluate the inherent capabilities of LLM.

**Weaknesses:**

* It is better to add the evaluation of code generation to the general abilities.

**Questions:**

* Why evaluate on the LLaMA-2-Chat instead of LLaMA-2? Could you provide the results for LLAMA-2 as well?

* For Tables 8 and 9, could you provide the results of all tasks at each round?

---

> ### Author Response · Authors · 2023-11-21
> **Official Comment by Authors**
>
> Thank you for taking the time to review our paper and for providing feedback.
>
> We kindly request a few minutes of your time to carefully consider our clarifications and responses.
>
> > **W1**: It is better to add the evaluation of code generation to the general abilities.
>
> Following your advice, we have incorporated an assessment of code generation capabilities. We selected the MBPP dataset as the baseline for evaluating code abilities and conducted evaluations across six different models.
>
> Furthermore, to assess the applicability of our TRACE benchmark findings to larger models, we have included evaluations on the Llama2-70B-Chat model. These evaluations focused on the performance of sequential target tasks and changes in general capabilities. The results in Tables 1 and 2 demonstrate that larger models also suffer from significant catastrophic forgetting, echoing our findings with smaller models.
>
>
>
>
>
> > **Q1**: Why evaluate on the LLaMA-2-Chat instead of LLaMA-2? Could you provide the results for LLAMA-2 as well?
>
> Our choice is grounded in the practical consideration that **general users are more likely to seek enhancements on stronger models**. The Chat variant of LLaMA models, compared to the Base versions, exhibit superior capabilities in instruction following and safety, making them more suitable for real-world applications. In the context of continual learning, leveraging a model like LLaMA-2-Chat, which already possesses a strong baseline in crucial capabilities, allows us to explore the potential for creating a more robust and comprehensive model.
>
> Considering your suggestion, we have also included results for the base LLaMA-2-7B model for comparison. These can be found in Appendix, Table 10. The data indicates a noticeable performance gap between the two variants, with LLaMA-2-Chat demonstrating stronger instruction-following abilities and overall performance, validating our choice for its use in our experiments.
>
>
>
> > **Q2**: For Tables 8 and 9, could you provide the results of all tasks at each round?
>
> We have now updated  Tables 8 and 9 to include the performance of all tasks at each round.
>
> In the revised tables, the upper triangle represents the performance on tasks that have been trained, while the lower triangle shows the performance on unseen tasks. The experimental data clearly indicates that both llama2-7b-chat and llama2-13b-chat exhibit generally poor performance on the unseen tasks in TRACE. This highlights the challenging nature of our benchmark.

---

> ### Comment · Reviewer_UTqq · 2023-11-22
>
> Thank you for your response, which has addressed most of my concerns. Considering this, I have decided to maintain my initial scores.

---

### Official Review · Reviewer_aiNx · 2023-10-28

**Soundness:** 4 excellent
**Presentation:** 3 good
**Contribution:** 3 good
**Rating:** 5
**Confidence:** 3

**Summary:**

This paper identifies limitations of existing continual learning (CL) benchmarks when applying to instruction-tuned (aligned) LLMs and proposes a new CL benchmark, TRACE, designed for aligned LLMs. The TRACE benchmark consists of eight tasks to evaluate LLMs in domain-specific, multilingual, code, and mathematical reasoning abilities. In addition to traditional CL metrics, e.g., overall performance and forgetting, they also evaluate aligned LLMs' general ability, instruction-following ability, and safety ability after learning sequentially.

Experiments on Llama2, Vicuna, and Baichuan2 with 7B and 13B sizes show that nearly all models exhibit a significant decline in general abilities, instruction-following ability, and math and reasoning ability after continual learning the TRACE benchmark. The author further proposes a mitigation method, called Reasoning-augmented Continual Learning (RCL), to encourage the model to generate task analyses and rationales during training, which delivers strong performance on target tasks and substantially retains the original strengths of LLMs.

Overall, this paper is well-written with solid experiments, ablation, and analysis. However, the evaluated CL baselines are limited, leaving the effectiveness of other CL methods in this context unknown.

**Strengths:**

- The biggest contribution of this paper is providing a CL benchmark designed for evaluating aligned LLMs. Evaluating aligned LLMs' general, instruction-following, and safety abilities after learning sequentially is novel. Some findings are interesting; for example, using LoRA for CL will have a more negative impact on the instruction-following ability of LLMs.

- This paper is well-written and easy-to-follow. The experiments are solid with detailed ablation and analysis, including the number of training data, epochs and base models.

**Weaknesses:**

- The compared CL baselines are limited. While O-Lora and PP are used, only Replay and Lora is used in the main experiments. The effectiveness of other CL methods in the context of continually learning instruction-tuned LLMs is unknown. For example, the traditional EWC, GEM, and recently L2P [1], DynaInst [2]. Can these methods reduce forgetting while maintaining the general, instruction-following, and safety ability?
- For instruction-tuned LLMs, generalisation ability on unseen tasks is also essential as it may reduce the training data needed for learning new tasks [3][4]. It is good to have a forward transfer (FWT) metric as well.

[1] Learning to Prompt for Continual Learning, CVPR 2022

[2] Large-scale Lifelong Learning of In-context Instructions and How to Tackle It, ACL 2023

[3] CITB: A Benchmark for Continual Instruction Tuning, EMNLP 2023

[4] Super-NaturalInstructions: Generalization via Declarative Instructions on 1600+ NLP Tasks, EMNLP 2022

- Typos:
1.	P3 Eq2, is it a typo for i≥t? If OP is to measure the overall performance of currently learned t tasks, shouldn’t it be i≤t?
2.	typo for Table.2, Vicuna-13B-V1.5 $\Delta R^G_t$ shouldn’t be 0?
3.	P8 Table.6 → Figure.6
4.	P9 Table.6 → Table.4
5.	P23 Table.26, BWT is ???

**Questions:**

1.	I’m not sure why in-context learning (ICL) can be a CL baseline. What’s the meaning of using ICL as a CL baseline?
2.	Table 1, why does ICL not have BWT?
3.	For SeqLoraFT, are you initialising a new adapter for learning each new task or using the same adapter?
4.	For RCL, for non-reasoning tasks such as Py150, how do you generate the reasoning steps? Is there any example?
5.	Task order: which CL method does Table 26 use? Is there any analysis of the impact of task ordering?

---

> ### Author Response · Authors · 2023-11-21
> **Official Comment by Authors (Part 1: Weaknesses）**
>
> Thank you for your constructive feedback on our paper. We truly appreciate the time you invested in the review. We've carefully considered your insights and addressed the highlighted concerns. We hope our responses shed more light on the matter.
>
> > **W1**: The compared CL baselines are limited. While O-Lora and PP are used, only Replay and Lora is used in the main experiments. The effectiveness of other CL methods in the context of continually learning instruction-tuned LLMs is unknown. For example, the traditional EWC, GEM, and recently L2P [1], DynaInst [2]. Can these methods reduce forgetting while maintaining the general, instruction-following, and safety ability?
>
> Thank you for your suggestions. Actually, traditional continual learning methods can face many challenges when using LLMs. For example, for regularization-based approaches such as EWC, GEM and OGD, etc.,  these methods require storing gradients or parameters from previous tasks, which poses great challenges to computation and storage capabilities and are often practically infeasible due to severe hardware limitations. Architecture-based methods (such as L2P and PP), their requirement for task-ID can be impractical in many real-world scenarios. Besides, the DynaInst method can be categorized into the replay-based method and it is hard for us to reproduce due to the lack of code in its paper.
>
> We have added detailed descriptions for the limitations of these methods in the related work (Section 2.1).
>
>
>
> > **W2**: For instruction-tuned LLMs, generalisation ability on unseen tasks is also essential as it may reduce the training data needed for learning new tasks 3. It is good to have a forward transfer (FWT) metric as well.
>
> Thank you for your valuable suggestion regarding the inclusion of unseen task testing to evaluate the generalization ability of instruction-tuned LLMs. We agree with the importance of this aspect and have made the following additions to our paper:
>
> 1. **Related Work Comparison**: We have expanded the related work section to include a comparison with the study referenced in [3]. Unlike [3], which assesses the performance of instruction tuning on a base model, TRACE is tailored for aligned models, which typically undergo extensive instruction tuning on large datasets. This presents unique challenges in constructing an unseen task set.
> 2. **Performance on Unseen Tasks:** To address this, we evaluated the performance of LLMs on unseen tasks within TRACE. The table below presents the performance of llama2-7b-chat across each round on both current and unseen tasks, illustrating generally low effectiveness on most tasks with no significant performance change.
>
> | task\round  | 1    | 2    | 3    | 4    | 5    | 6    | 7    | 8    |
> | ----------- | ---- | ---- | ---- | ---- | ---- | ---- | ---- | ---- |
> | C-STANCE    | 0.61 |      |      |      |      |      |      |      |
> | FOMC        | 0    | 0.62 |      |      |      |      |      |      |
> | MeetingBank | 0.07 | 0    | 0.5  |      |      |      |      |      |
> | Py150       | 0.06 | 0.05 | 0.05 | 0.56 |      |      |      |      |
> | ScienceQA   | 0    | 0.06 | 0    | 0    | 0.72 |      |      |      |
> | NumGLUE-cm  | 0    | 0.02 | 0    | 0    | 0.33 | 0.69 |      |      |
> | NumGLUE-ds  | 0    | 0.03 | 0    | 0    | 0.14 | 0.15 | 0.57 |      |
> | 20Minuten   | 0.39 | 0.37 | 0.36 | 0.39 | 0.36 | 0.36 | 0.37 | 0.4  |
>
> 3. **Forward Transfer Metrics**: We have added forward transfer metrics for both baseline methods and our RCL approach. Detailed results are now included in our revised Table 1 and Table 3.
>
>
>
> > **Typos**
>
> Thank you for pointing out these typos in our paper. We have thoroughly reviewed and corrected them.
>
> We apologize for the oversight in the initial version and appreciate your valuable review.

---

> ### Author Response · Authors · 2023-11-21
> **Official Comment by Authors (Part 2: Questions）**
>
> > **Q1**: I’m not sure why in-context learning (ICL) can be a CL baseline. What’s the meaning of using ICL as a CL baseline?
>
> We employed In-context Learning (ICL) as a benchmark in our study to gauge the complexity of tasks in TRACE. The logic is straightforward: if ICL, without additional training, can efficiently handle these tasks, it implies that TRACE is not challenging enough, thereby eliminating the need for continual training and concerns of catastrophic forgetting.
>
>
>
>
>
> > **Q2**: Table 1, why does ICL not have BWT?
>
> ICL doesn't have Backward Transfer (BWT) as it's not a continual learning method and doesn't involve model training. Therefore, metrics like BWT and FWT are not applicable and are omitted from ICL in our analysis.
>
>
>
> > **Q3**: For SeqLoraFT, are you initialising a new adapter for learning each new task or using the same adapter?
>
> In our SeqLoraFT experiments, we utilize the same adapter across all tasks. This decision is rooted in two main considerations:
>
> **Preserving the One-for-All Utility of** **LLMs**: Creating a new adapter for each task would essentially equate to learning a separate model for each task, deviating from the current versatile application of LLMs. Using a single adapter aligns with the LLM's inherent capability to handle a multitude of tasks without the need for distinct models.
>
> **Enhanced Generalization to Unseen Tasks**: If separate adapters were created for each task, it would inherently limit the model's ability to adapt and perform well on tasks it has not been explicitly trained for. By using the same adapter, we aim to cultivate a model that retains its adaptability and effectiveness across a wider range of tasks, including those it has not encountered before.
>
>
>
> > **Q4**: For RCL, for non-reasoning tasks such as Py150, how do you generate the reasoning steps? Is there any example?
>
> In RCL, for tasks not inherently based on reasoning, we begin by dissecting the problem to identify key relationships. This approach simplifies the response generation process for the model. Specifically for Py150, a code generation task, we employ code comments as a medium for outlining problem analysis and reasoning steps, such as:
>
> ```
> [prompt]
>
> Complete the next line of the following codes. Give your reasoning as annotation first, and then answer.
>
> from flask import request , session <EOL> from flask import Blueprint <EOL> import api <EOL> import json <EOL> import mimetypes <EOL> import os . path <EOL> import api . auth <EOL> import asyncio <EOL> import threading <EOL> from api . annotations import api_wrapper <EOL> from api . common import flat_multi <EOL> from api . exceptions import * <EOL>
>
> [answer]
>
> // The code imports various modules and functions from Flask, a web framework for Python. A Blueprint object is a way to organize a group of related views and other code. It is used for creating a modular application. Hence, the next line of code creates a Blueprint object with a specific name and the current module as the Blueprint's package. <EOL>
>
> blueprint = Blueprint ( \"<STR_LIT>\" , **name** )
> ```
>
>
>
>
>
> > **Q5**: Task order: which CL method does Table 26 use? Is there any analysis of the impact of task ordering?
>
> In the original version of our paper, Table 26 (now Table 28 in the revised paper) presents the performance of LLaMA-7b-chat without the application of any continual learning methods, specifically using task order 2.
>
> It is important to note that there's a significant difference in performance between task orders. While the overall performance for the original order is 48.7, the average performance for order 2 drops to 32.9, indicating that **the sequence of tasks has a substantial impact on the model's final performance.**

---

### Official Review · Reviewer_hYjF · 2023-10-31

**Soundness:** 2 fair
**Presentation:** 1 poor
**Contribution:** 2 fair
**Rating:** 3
**Confidence:** 5

**Summary:**

The paper presents a new benchmark for continual learning of aligned LLMs. To this end, the author has mixed existing challenging benchmarks in a sequential manner, where these benchmarks are not used for recently published aligned LLMs, e.g., Lamma2-chat. The authors also present a new method called reasoning-augmented continual learning which augments the reason generated from GPT4.

**Strengths:**

1. The authors have considered somewhat large language models for the evaluation. It would be very grateful if the authors could compare it on a bigger scale, e.g., llama2 70b-chat, falcon-40b instruct.

**Weaknesses:**

1. While I do agree that some papers, e.g., instruction tuning, have used the past continual learning benchmarks, TRACE also shares the same issue. After publishing this benchmark, there are possibilities that some may use this dataset for pre-training. So, I think continual learning (CL) benchmarks all shares a similar problem, so this is hard to claim as a problem of prior CL benchmark (since this paper also have the same issue).

2. For me, it is hard to understand i) why this benchmark has novelty for CL and 2) why this has novelty for alignment LLMs
- i) TRACE is a mix of existing challenging benchmarks and put them in a sequential manner. It is hard to see novelties or specialization for continual learning (except they have put it in a sequential manner).
- ii) The only thing this benchmark is specialized to alignment LLMs is that they propose a metric that is related to alignment LLMs. But these metrics are also a naive extension of prior continual learning metrics (the overall performance and backward transfer suggested by [1]).

3. One important discussion in continual learning is about forward transfer [1,2,3]. Considering forward transfer metric is also needed.

4. As a benchmark paper, I think it is very important to compare the prior CL methods, where this paper only shows a few baselines. For instance, considering the following methods will be helpful (see some baselines in [4]) or consider [5,6,7].

5. The overall writing should be improved. For instance, **the main text cites the results in the Appendix too much** (Figure 5, Table 6). If the table and figures are important, the authors should put them in the main text. Especially, the results of reasoning-augmented continual learning are important but missing in the main text.

6. Using GPT-4 is not a good choice for continual learning (even for reasoning), since we don't know which dataset they have used to train the model.

Overall, it is somewhat hard to find a novelty as a continual learning benchmark (and typically specialized for aligned LLMs), and it is hard to find novel metrics. Furthermore, the paper needs more consideration regarding forward transfer, more comparison of multiple frameworks, and improving the overall writing.

Please find the reference below\
[1] Lopez-Paz and Ranzato, Gradient episodic memory for continual learning, NeurIPS 2017\
[2] Wolczyk et al., Continual World: A Robotic Benchmark For Continual Reinforcement Learning, NeurIPS 2021\
[3] Chen et al., Is Forgetting Less a Good Inductive Bias for Forward Transfer? ICLR 2023\
[4] Jang et al., Towards Continual Knowledge Learning of Language Models, ICLR 2022\
[5] D'Autume et al., Episodic memory in lifelong language learning, NeurIPS 2019\
[6] Huang et al., Continual learning for text classification with information disentanglement based regularization, NACCL 2021\
[7] Razdaibiedina et al., Progressive prompts: Continual learning for language models, ICLR 2023

**Questions:**

I think the result of Vicuna-13B-V1.5 in Table 2 should be 0.

---

> ### Author Response · Authors · 2023-11-21
> **Official Comment by Authors (Part 1: Novelty）**
>
> We appreciate your feedback and have carefully considered your suggestions. We try to address your remarks below:
>
> > **W2**: For me, it is hard to understand i) why this benchmark has novelty for CL and 2) why this has novelty for alignment LLMs
> >
> > - i) TRACE is a mix of existing challenging benchmarks and put them in a sequential manner. It is hard to see novelties or specialization for continual learning (except they have put it in a sequential manner).
> > - ii) The only thing this benchmark is specialized to alignment LLMs is that they propose a metric that is related to alignment LLMs. But these metrics are also a naive extension of prior continual learning metrics (the overall performance and backward transfer suggested by [1]).
>
> Regarding your concerns about the novelty of TRACE. We'd like to provide the following clarifications:
>
> a. **Unique Design of TRACE**: As detailed in Section 4.1, our innovation lies not merely in the datasets used, but significantly in the design of TRACE. **We ensured the benchmark's high challenge level and task diversity to comprehensively cover abilities essential for enhancement in** **LLMs**. After extensive experimentation with numerous datasets, we strategically selected these eight datasets for their optimal combination of challenge and diversity.
>
> b. **Metrics Beyond Complexity**: Our focus on metrics isn't about their complexity but about **highlighting crucial aspects often overlooked in** **LLM** **continual learning settings**. TRACE emphasizes not just sequential task performance, but also the general capabilities, instruction adherence, and safety dynamics of LLMs.
>
>
>
> In addition to our previous points regarding the design of TRACE and our novel metrics, we'd like to highlight a key contribution of our work:
>
> c. **Systematic Validation Across Diverse Models**: A significant aspect of our contribution lies in the extensive validation of TRACE across six models, ranging from 7B to 70B in size. Our experiments were not limited to the evaluation of eight sequential target tasks but also included a systematic assessment across six dimensions of general capabilities. The findings demonstrate that catastrophic forgetting is a pervasive issue across models of varying sizes, from 7B to 70B. **These results are crucial as they provide valuable insights for future research in continual learning of LLMs**, indicating that the problem of forgetting is not confined to a specific model size but is a broader challenge that needs addressing.

---

> ### Author Response · Authors · 2023-11-21
> **Official Comment by Authors (Part 2: Clarification）**
>
> > **W1**: While I do agree that some papers, e.g., instruction tuning, have used the past continual learning benchmarks, TRACE also shares the same issue. After publishing this benchmark, there are possibilities that some may use this dataset for pre-training. So, I think continual learning (CL) benchmarks all shares a similar problem, so this is hard to claim as a problem of prior CL benchmark (since this paper also have the same issue).
>
> Thank you for your insightful comment regarding the potential for future pre-training on our TRACE benchmark.
>
> Indeed, **our primary intention was to design a benchmark that effectively evaluates the extent and presence of catastrophic forgetting in** **Large Language Models**. While we acknowledge the risk of our dataset eventually being used for pre-training, thus potentially encountering similar challenges as existing benchmarks, our core findings regarding the significant forgetting in current LLMs remain valid and generalizable.
>
> To address your concern, we have revised the Introduction and Section 2.2 of our paper to explicitly acknowledge this potential limitation.
>
>
>
> > **W3**: One important discussion in continual learning is about forward transfer [1,2,3]. Considering forward transfer metric is also needed.
>
> Following your suggestion, we have now included a detailed discussion and evaluation of forward transfer in our revised paper.
>
> Specifically, you can find the forward transfer scores for baseline methods in Table 1. Additionally, the results demonstrating the forward transfer efficacy of our Reasoning-augmented Continual Learning (RCL) method are presented in Table 3.
>
>
>
> > **W4**: As a benchmark paper, I think it is very important to compare the prior CL methods, where this paper only shows a few baselines. For instance, considering the following methods will be helpful (see some baselines in [4]) or consider [5,6,7].
>
> Thanks for your suggestions. First, we have conducted [7] (PP) as a baseline. Second, traditional continual methods can be infeasible when faced with models with huge amounts of parameters. For example,  for regularization-based approaches such as EWC, GEM and OGD, etc.,  these methods require storing gradients or parameters from previous tasks, which poses great challenges to computation and storage capabilities and are often practically infeasible due to severe hardware limitations.
>
> For [5] you have mentioned, the method can be categorized into replay-based methods and the "retrieval then inference" may lead to huge computation overhead. For [6], the IDBR method was primarily developed for classification tasks and can not be directly applied to text-to-text tasks.
>
> We have added detailed descriptions for the limitations of these methods in the related work (Section 2.1).

---

> ### Author Response · Authors · 2023-11-21
> **Official Comment by Authors (Part 3: Clarification++）**
>
> > **W5**: The overall writing should be improved. For instance, **the main text cites the results in the Appendix too much** (Figure 5, Table 6). If the table and figures are important, the authors should put them in the main text. Especially, the results of reasoning-augmented continual learning are important but missing in the main text.
>
> We apologize for any inconvenience caused by the previous layout of our paper, which excessively referenced the Appendix for critical results. The extensive scope of experiments conducted with TRACE required careful consideration to fit within the page limit, hence the initial placement of figures and tables in the Appendix.
>
> **We have taken your feedback seriously and, in our revised version, we have streamlined the content and reorganized the paper based on the significance of the results.** We have managed to condense the paper from 66 pages to 25 pages, ensuring that key findings are now prominently featured in the main text.
>
>
>
> > **W6**: Using GPT-4 is not a good choice for continual learning (even for reasoning), since we don't know which dataset they have used to train the model.
>
> We appreciate your concern about the suitability of using GPT-4 for continual learning and reasoning tasks due to the unknown specifics of its training datasets. In response to this point:
>
> 1. **Model Version and Data Cut-off**: The version of GPT-4 employed in our study is the 0311 variant, which is updated only up to the year 2021. This aspect is pertinent since all datasets included in TRACE were released post this date, ensuring that they were not part of GPT-4's training data.
> 2. **Reasoning Path** **Validation**: To ensure the validity of the reasoning paths generated by GPT-4, we conducted a thorough check comparing its outputs against the ground truths of the TRACE datasets.  As detailed in Section 5, the results showed a high correctness rate of 94%, indicating the robustness of GPT-4’s reasoning capabilities suitable for our study.
>
> Our results indicate that GPT-4, despite the lack of transparency in its training datasets, remains a competent model for generating reasoning paths in the context of our TRACE benchmark.
>
>
>
> > **Q1**: I think the result of Vicuna-13B-V1.5 in Table 2 should be 0.
>
> Thank you for pointing out the error in the reported result for Vicuna-13B-V1.5 in Table 2. We have corrected this error in the revised version of our paper. In addition, we have conducted a thorough re-check of all other experimental results to ensure their accuracy and reliability.

---

### Official Review · Reviewer_dqAr · 2023-10-31

**Soundness:** 3 good
**Presentation:** 3 good
**Contribution:** 3 good
**Rating:** 6
**Confidence:** 4

**Summary:**

This paper proposes an evaluation benchmark for continual learning of LLMs. In addition, RCL was proposed for reducing the loss of general abilities of LLMs during tuning. The proposed benchmark consists of carefully curated data from different domains.

**Strengths:**

1. A comprehensive benchmark and in-depth analysis of LLaMA-2, Vicuna, Baichuan2 is provided, revealing the observation of catastrophic forgetting of (relatively small scale) LLMs when tuning on novel tasks.
2. Reasoning-based data is found important to mitigate forgetting of LLMs in tuning.

**Weaknesses:**

1. Although tuning larger models would be computationally expensive, it would be still interesting to see whether the observation and analysis hold with more parameters, since larger models may behave differently.
2. The proposed reasoning-augmented method indeed provides insights for tuning LLMs, however, the method and study of it are too simple. For example, currently, it involves the manual selection of GPT-4 augmented data. Also it would be interesting to see how many reasoning-augmented data would be "optimal" for avoiding forgetting. Does more data help or hurt?

**Questions:**

1. Can you provide more explanation as to why reasoning-augmented data is helpful for continual learning of LLMs?

---

> ### Author Response · Authors · 2023-11-21
> **Official Comment by Authors**
>
> Thank you for your appreciation and an excellent summarization of our work.  We appreciate your time and effort in carefully reviewing our work. We try to address your remarks below:
>
> > **W1**: Although tuning larger models would be computationally expensive, it would be still interesting to see whether the observation and analysis hold with more parameters, since larger models may behave differently.
>
> Thank you for suggesting the evaluation of larger models to determine if our observations and analyses are consistent across different scales. In response, we have extended our study to include assessments of the larger Llama2-70B-Chat model. Our findings, which are detailed in Tables 1 and 2 of our revised paper, offer intriguing insights:
>
> **Superior Performance of Larger Models**: The performance of the 70B-sized Llama2-chat model is significantly better than its 13B and 7B counterparts, particularly in the context of final task performance.
>
> **Catastrophic Forgetting in Larger Models**: Despite its enhanced capabilities, the 70B Llama2-chat model still exhibits notable catastrophic forgetting. For instance, in sequential target tasks, the LoRASeqFT method showed a backward transfer score of -17.5%, and in general capabilities, a score of -10.4%.
>
> **Replay Method Across Sizes**: When using the replay method, the final performance on target tasks is relatively similar across different sizes of Llama2-chat models. However, the preservation of general capabilities is more robust in larger models.
>
>
>
> > **W2**: The proposed reasoning-augmented method indeed provides insights for tuning LLMs, however,**the method and study of it are too simple.** For example, currently, it involves the manual selection of GPT-4 augmented data. Also it would be interesting to see how many reasoning-augmented data would be "optimal" for avoiding forgetting. Does more data help or hurt?
>
> Thank you for your constructive comments on our RCL approach. I’d like to provide the following clarifications and updates:
>
> **Methodology Correction**: We acknowledge an error in our initial description regarding the use of GPT-4 augmented data. The process involved manual evaluation of select demonstration examples, not every data point. This has been corrected in Section 5 to state: "Reasoning was verified against ground truth".
>
> **Experimentation with Varied Data Sizes**: In response to your query about the optimal amount of reasoning-augmented data for mitigating forgetting, we have expanded our empirical analysis. The revised Table 3 in our paper now includes results from experiments with varying data sizes. **These findings illustrate a notable performance improvement in RCL when the training dataset size increases from 0.5k to 5k samples, with the final performance metric rising from 46.6 to 51.2**.
>
> **Balancing Performance and Forgetting**: Despite the advantages of larger datasets in task-specific performance, they also exacerbate the issue of forgetting. Our analysis suggests that a dataset size of 0.5k strikes an effective balance between maintaining the original capabilities of LLMs and optimizing performance on new tasks.
>
> **Inclusion of Forward Transfer Metrics**: Alongside overall and backward transfer metrics, we have now incorporated forward transfer scores for a more rounded evaluation of our approach.
>
>
>
> > **Q1**: Can you provide more explanation as to why reasoning-augmented data is helpful for continual learning of LLMs?
>
> In response to your query about the effectiveness of reasoning-augmented data in continual learning for Large Language Models , our approach addresses two key aspects:
>
> a. **Prevention of Shortcut Learning**: In continual learning scenarios, LLMs might develop a propensity for shortcut learning [1], where they focus only on superficial features or patterns of a problem. This approach neglects the deeper logic and underlying relationships essential for a thorough understanding of tasks. Reasoning-augmented data counters this tendency by compelling the models to engage with the core aspects of tasks, fostering a deeper and more comprehensive learning process [2][3].
>
> b. **Maintenance of Generalization Abilities**: The use of reasoning-augmented data encourages LLMs to leverage their existing knowledge and reasoning abilities when learning new tasks. This method enhances the models' generalization abilities as they learn to apply known knowledge and reasoning skills to new or unfamiliar tasks. This adaptability and broad applicability are vital for LLMs to remain effective in the rapidly evolving field of language processing.
>
>
>
> **Reference**
>
> [1] Shortcut learning in deep neural networks
>
> [2] Explanations from large language models make small reasoners better
>
> [3] Chain-of-Thought Prompting Elicits Reasoning in Large Language Models

---

> > ### Comment · Reviewer_dqAr · 2023-11-22
> >
> > Thanks for the response from the authors. Most of my concerns have been addressed and I have increased the rating.

---

### Author Response · Authors · 2023-11-21
**Overall response to reviewers**

We greatly appreciate your thorough reviews and constructive feedback. In response, we have undertaken an extensive revision of our paper, investing substantial effort to address each of your insightful comments and suggestions. These revisions are not just extensive in scope but also intensive in their depth, reflecting our commitment to enhancing the paper's quality and rigor. Detailed below are the key areas where we have focused our considerable revision efforts:

1. **Comprehensive Evaluation of Catastrophic Forgetting in** **LLMs**:
   1. [suggested by Reviewer dqAr]  **Expanded Model Evaluation**: We have extended our investigation of catastrophic forgetting to larger models, notably incorporating an evaluation of the LLaMA2-70B-Chat model.
   2. [suggested by Reviewer UTqq]  **Enhanced General Capability Assessment**: Following Reviewer 3’s advice, we added an evaluation dimension for code generation capabilities to more comprehensively assess the general abilities of LLMs.
2. **Enrichment of Experimental Data and Analysis**:
   1. [suggested by Reviewer dqAr]  **Dataset and Method Expansion**: We expanded our Reasoning Augmented dataset size (from 0.5k * 8 to 5k * 8) and included additional experiments across various sizes using our RCL (Reasoning-augmented Continual Learning) method.
   2. [suggested by Reviewer hYjF and Reviewer aiNx]  **Inclusion of Forward Transfer Metrics**: Both our main experiments and RCL methodology now incorporate results for the forward transfer (FWT) metric.
   3. [suggested by Reviewer UTqq]  **Continual Learning Experiment with LLaMA2-Base**: Additional continual learning experiments with LLaMA2-Base have been conducted, with results detailed in Appendix Table 10.
3. **Content Refinement and Clarification**:
   1. [suggested by Reviewer hYjF and Reviewer aiNx]  **Analysis of Traditional Continual Learning Methods on LLMs**: We added a paragraph in the Related Work section, particularly focusing on the challenges of applying traditional continual learning methods to LLMs.
   2. [suggested by Reviewer hYjF]  **Acknowledgement of TRACE's Potential Limitation**s: The Related Work section has been augmented with a discussion on the potential limitations of TRACE.
   3. [suggested by Reviewer hYjF]  **Article Reorganization and Reduction**: Based on content significance, we have restructured the manuscript and eliminated redundant appendices, effectively reducing the paper from 66 to 25 pages.

[suggested by Reviewer hYjF and aiNx] Additionally, we have meticulously reviewed and corrected typos throughout the manuscript and rigorously verified experimental results for accuracy.


We have uploaded the code for TRACE in the supplementary materials.